# The Genomic-Driven Discovery of Glutarimide-Containing Derivatives from *Burkholderia gladioli*

**DOI:** 10.3390/molecules28196937

**Published:** 2023-10-05

**Authors:** Hanna Chen, Xianping Bai, Tao Sun, Xingyan Wang, Youming Zhang, Xiaoying Bian, Haibo Zhou

**Affiliations:** 1Helmholtz International Lab for Anti-Infectives, Shandong University–Helmholtz Institute of Biotechnology, State Key Laboratory of Microbial Technology, Shandong University, Qingdao 266237, China; chenhannahappy@163.com (H.C.); xpbai201812479@163.com (X.B.); suntao_go@126.com (T.S.); 202012620@mail.sdu.edu.cn (X.W.); 2School of Medicine, Linyi University, Shuangling Road, Linyi 276000, China; 3CAS Key Laboratory of Quantitative Engineering Biology, Shenzhen Institute of Synthetic Biology, Shenzhen Institute of Advanced Technology, Chinese Academy of Sciences, Shenzhen 518055, China

**Keywords:** glutarimide-containing polyketides, genome mining, *Burkholderia gladioli*, heterologous expression, promoter engineering

## Abstract

Glutarimide-containing polyketides exhibiting potent antitumor and antimicrobial activities were encoded via conserved module blocks in various strains that favor the genomic mining of these family compounds. The bioinformatic analysis of the genome of *Burkholderia gladioli* ATCC 10248 showed a silent *trans*-AT PKS biosynthetic gene cluster (BGC) on chromosome 2 (Chr2C8), which was predicted to produce new glutarimide-containing derivatives. Then, the silent polyketide synthase gene cluster was successfully activated via in situ promoter insertion and heterologous expression. As a result, seven glutarimide-containing analogs, including five new ones, gladiofungins D-H (**3**–**7**), and two known gladiofungin A/gladiostatin (**1**) and **2** (named gladiofungin C), were isolated from the fermentation of the activated mutant. Their structures were elucidated through the analysis of HR-ESI-MS and NMR spectroscopy. The structural diversities of gladiofungins may be due to the degradation of the butenolide group in gladiofungin A (**1**) during the fermentation and extraction process. Bioactivity screening showed that **2** and **4** had moderate anti-inflammatory activities. Thus, genome mining combined with promoter engineering and heterologous expression were proved to be effective strategies for the pathway-specific activation of the silent BGCs for the directional discovery of new natural products.

## 1. Introduction

Glutarimide-containing polyketides (PKs), well known for antitumor and antifungal properties, are generally isolated from *Streptomyces* species, including lactimidomycin (LTM) [1,2], migrastatin (MGS) [3,4], iso-migrastatin (iso-MGS) [4], cycloheximide [5], and streptimideones [6,7,8]. In recent years, glutarimide antibiotics were discovered one by one in various strains. For example, the Medema group found the glutarimide modules in various gammaproteobacterial pathogens, *Burkholderia gladioli* and *Pseudomonas* spp., as well as *Magnetospirillum* and *Geobacter*, based on the *trans*-acyltransferase polyketide synthase (*trans*-AT PKS) Annotation and Comparison Tool (transPACT), to understand the evolution of combinatorial diversity in *trans*-AT PKS assembly [9]. Hertweck et al. and Challis et al. almost simultaneously discovered the same glutarimide antibiotic, named gladiofungin and gladiostatin, respectively, from the Gram-negative strain *B. gladioli* using genome mining [10,11]. The other glutarimide family polyketides, secimide from *P. syringae* pv. *syringae* and sesbanimides, biosynthesized by marine alphaproteobacterial strains, were also discovered [9,12,13] (Appendix A).

These glutarimide-containing antibiotics were biosynthesized via *trans*-AT PKSs, which consist of multiple modules harboring various functional domains. Each module usually at least contains a ketosynthase (KS) and an acyl carrier protein (ACP), and other additional domains, such as dehydratase (DH) domain, enoyl reductase (ER) domain, and keto reductase (KR) domain [14,15,16]. The outstanding modification of *trans*-AT polyketides is the formation of the β-branch, which is the conversion of a β-carbonyl function into the carbon skeleton [17]. Moreover, one or more free-standing *trans*-ATs in the biosynthesis of polyketides select coenzyme A (CoA)-bound acyl building blocks and transfer them onto ACP domains, and the release of polyketide backbone is always catalyzed by a thioesterase (TE) domain [14,15,16]. The conserved glutarimide moiety of glutarimide antibiotics was generated by three conserved genes encoding AT, ACP, an asparagine synthetase homolog, combined with the seven conserved domains of the original module (KS-DH-KR-ACP-KS-B-ACP) mediating a Michael-type vinylogous addition of a malonyl unit followed by a cyclization [5,18,19] (Appendix A). Compared to the glutarimide-containing polyketides from *Streptomyces* species and the others, gladiofungins and gladiostatin have an unusual butenolide moiety, which is derived from a special polyketide chain-release reaction catalyzed by an A-factor synthase (AfsA)-like offloading domain [10,11] (Figure 1 and Appendix A). This chain-release mechanism led to the significant structural differentiation of gladiofungins and gladiostatin from other glutarimide antibiotics. Additionally, this *trans*-AT biosynthetic gene cluster (BGC) is highly conserved in diverse *B. gladioli* strains, including mushroom, plant, and human pathogens [10].

At present, the two main efficient pathway-specific strategies for genome-mining natural products are in situ activation and heterologous expression [20,21,22]. Recently, our group successfully activated several silent/cryptic BGCs in *Burkholderiales* strains via in situ modifications using efficient recombination systems [23,24,25]. The heterologous expression of BGCs is not only an effective approach to optimizing the production yield of valuable biomolecules but also an efficient strategy to mine silent/cryptic secondary metabolites in the post-genomic era [26,27]. The strain *Caldimonas brevitalea* DSM 7029 (previously named *Schlegella brevitalea*) was used as a heterologous host to produce Myxobacterial and Burkholderial secondary metabolites, such as epothilone, chitinimides, rhizomide, etc. [28,29]. The Gram-negative model bacterium *B. thailandensis* E264 harbors an abundance of BGCs (23), which produces nonribosomal peptides (NRPs), PKs, bacteriocins, terpenes, various hybrid molecules, etc. [30,31]. This strain possesses four essential 4′-phosphopantetheinyl transferases for the efficient biosynthesis of PKs and NRPs and exhibits an advantageous chassis for the production of natural products from Gram-negative proteobacteria [32,33].

In this study, we successfully activated a silent *trans*-AT PKS BGC (*gla*) that potentially produces new glutarimide-containing derivatives through the in situ insertion of the constitutive promoter and heterologous expression in *C. brevitalea* DSM 7029 and *B. thailandensis* E264. Seven corresponding glutarimide-containing polyketides (**1**–**7**) were elucidated, including five new ones, and compounds **2** and **4** had moderate anti-inflammatory activities. Therefore, the in situ insertion of a promoter and heterologous expression are effective methods for mining the silent *gla* BGC for the directional discovery of new glutarimide-containing natural products.

## 2. Results

### 2.1. Characterization of the Silent trans-AT Biosynthetic Gene Cluster

Bioinformatic analysis of the genome of *B. gladioli* ATCC 10248 based on the antiSMASH website (https://antismash.secondarymetabolites.org/upload/bacteria-1c2f0788-da11-4d64-9fc2-b207c22fee46/index.html#r1c8; accessed on 3 October 2023) showed a *trans*-AT PKS BGC8 on chromosome 2 (Chr2C8) with 94% similarity to the gladiostatin/gladiofungin BGC [10,11,23,34]. The assembly line of the Chr2C8, defined as *gla* BGC, had a noncanonical chain release domain AfsA predicted by the Pfam website (http://pfam-legacy.xfam.org/search/sequence; accessed on 3 October 2023) [35] compared to the canonical TE domain of PKS, which attracted us to investigate it (Figure 1). The *gla* PKS BGC contains nine core genes, *glaA-G* and *glaP* (Appendix A). The *glaA*, *glaB*, and *glaC* genes are responsible for the formation of the malonamyl thioester starter unit [10]. GlaD contains a seven-domain architecture to assemble the glutarimide group identical to the glutarimide moiety in *Streptomyces* (Figure 1). The genes *glaE* and *glaE1* encode the remaining eight modules. The assembly line has a noncanonical chain release domain AfsA that was different from the canonical TE domain in the *Streptomyces* and the other glutarimide assembly lines (Appendix A). This AfsA domain is responsible for releasing β-keto thioester from the upstream ACP domain by catalyzing the condensation of β-keto thioester with dihydroxyacetone phosphate (DHAP) [11]. The genes *glaP* and *glaG* encode a HAD family phosphatase and an NADP-dependent oxidoreductase, respectively. The upstream and downstream genes of *gla* BGC were transposase genes [36,37], indicating this *trans*-AT BGC might be acquired from horizontal gene transfer (HGT).

In the course of mining the novel products from the wild-type strain ATCC 10248 using the one strain many compounds (OSMAC) strategy [38], we did not detect the corresponding products of Chr2C8 from its crude extracts via LC-MS analysis, indicating that Chr2C8 was silent in our applied conditions. Then, we employed the insertion of the constitutive promoter P_Genta_ replacing the original promoter of the Chr2C8 using an efficient recombineering system Redγ-BAS from *P. aeruginosa* phage Ab31 in the ATCC 10248 [24] (Figure 2a). The inactivated mutant of the Chr2C8 was constructed by the gentamicin resistance gene, replacing the *glaC* gene (Figure 2a). According to our previous study, we deleted the gladiolin BGC in the two mutants ATCC 10248∆Chr2C8 and ATCC 10248P_Genta_-Chr2C8 through homologous recombineering in order to facilitate the detection of different peaks of the crude extract from the activated mutant, the inactivated mutant of Chr2C8, and the wild-type ATCC 10248 [24]. Then, HPLC-MS analysis showed a series of different peaks in the Chr2C8 activated mutant compared with the wild-type strain ATCC 10248 and inactivated mutant (Figure 2b). Thus, the silent Chr2C8 was activated in situ.

### 2.2. Structural Elucidation and Bioactivities Assay of Compounds ***1***–***7***

In order to elucidate the structures of these compounds produced by the activated mutant ATCC 10248Δ*gbn*:P_Genta_-Chr2C8, large-scale fermentations and metabolite isolation were performed. From 8 L fermentation of ATCC 10248Δ*gbn*:P_Genta_-Chr2C8, seven compounds (**1**–**7**) were purified and elucidated.

Compounds **1** and **2** were determined to be gladiofungin A (or gladiostatin) and its degradation product, named gladiofungin C, via a comparison of its ^1^H and ^13^C NMR and HR-ESI-MS data with the *m*/*z* 504. 2606 [M-H]^−^ and *m*/*z* 406.2596 [M-H]^−^, which has a prominent fragment of 154.0508 [M-H]^−^, a feature for a glutarimide moiety consistent with the reported gladiofungin [10,11] (Figure 2c and Appendix A). Gladiofungin D (**3**) was obtained as a colorless oil with the molecular formula C_22_H_35_NO_6_, as determined via HR-ESI-MS measurement (*m*/*z* 408.2741 [M-H]^−^). The ^1^H and ^13^C NMR data of **3** (Table 1 and Table 2) were similar to those of **2**, except that one carboxyl group at C-19 (*δ*_C_ 178.4) in **3** replaced the acetyl group [*δ*_H_ 2.13 (3H, s, H-20); *δ*_C_ 30.0 (C-20) and 209.6 (C-19)] in **2**, which was connected to C-18 (*δ*_H/C_ 2.33/33.9), supported by the HMBC correlation between H-18 and C-19 (Figure 2c and Figure 3).

Gladiofungin E (**4**) and gladiofungin F (**5**) were isolated as colorless oils. Their molecular formulas, C_26_H_39_NO_5_ and C_26_H_39_NO_6_, were determined on the basis of the negative HR-ESI-MS at *m*/*z* 444.2744 ([M-H]^−^) and 460.2691 ([M-H]^−^), respectively (Appendix A). The major portion of the glutarimide backbone could be assembled through the interpretation of the 1D NMR data (Table 1 and Table 2) and 2D correlations (Figure 3). The HMBC cross-peaks between H-24 (*δ*_H_ 1.97) and C-20 (*δ*_C_ 107.6), C-21 (*δ*_C_ 120.5) and C-22 (*δ*_H/C_ 7.04/137.3), and H-22 and C-19 (*δ*_C_ 156.7) suggested the presence of a β-methylfuran moiety in **4**, which was attached to C-18 (*δ*_H/C_ 2.54/28.2) according to its HMBC correlations from H-20 (*δ*_H_ 5.83) to C-18. The characteristic signals at *δ*_H/C_ 9.52/193.6 (C-22, CH) and *δ*_H/C_ 6.37, 6.22/137.1 (C-24, CH_2_) in **5** indicated the presence of an aldehyde group and terminal alkene residue, and its location was confirmed by HMBC correlations from H-24 to C-20 (*δ*_C_ 41.7), C-21(*δ*_C_ 143.5), and from H-20 (*δ*_H_ 3.35) to C-19 (*δ*_C_ 206.9) (Figure 3).

Gladiofungin G (**6**) and gladiofungin H (**7**) were isolated as colorless oils with molecular formulas of C_26_H_41_NO_6_ and C_26_H_43_NO_6_, respectively, based on the HR-ESI-MS at *m*/*z* 462.2864 ([M-H]^−^) and 464.3017 ([M-H]^−^) (Appendix A). The 1D NMR data of **6** (Table 1 and Table 2) showed several similarities to those of **5**, especially the chemical shifts of C-1–C-20 and C-1′–C-2′. The different signals in **6** were deduced to form a propylene oxide, which were further supported by the HMBC correlations from H-22 (*δ*_H_ 3.37, 3.45) and H-24 (*δ*_H_ 1.19) to C-20 (*δ*_C_ 48.3), C-21 (*δ*_C_ 72.6). Compound **7** was determined to be a ring-opening product of **6** according to its molecular weight and confirmed by 2D NMR data analysis (Figure 3).

Compound **2** was reported to be the degradation product of **1** [11]. To verify whether compounds **3**–**7** were also the degradation products of compound **1**, we conducted a time-series experiment which showed that compound **1** was the dominant product before 36 h, and then it decreased and was accompanied by the increase of other compounds (**2–7**) with the fermentation continuing (Appendix A). Additionally, we used HPLC-MS analysis to check the stability of compound **1**, which was stored in methanol solvent after being purified. After being stored for 2 days, compounds **2**–**3** could be detected obviously, and when extended to eight days, other unknown compounds belonging to this family were also detected but could not assigned to be **4**–**7** (Appendix A). According to the result and our isolation experience, we speculate that **2**–**7** were quite possibly derived from the degradation of **1**, and the degradation of **1** was mainly mediated by H_2_O during the formation and isolation process. But we cannot exclude another possibility, which is that these compounds are early released products. We will isolate more related products to clarify the true mechanism in the future.

The biological activity screening of the purified compounds **2**–**7** showed that compounds **2** and **4** had significant anti-inflammatory activities for their inhibition of NO production in LPS−induced RAW 264.7 macrophages; the others were not [39] (Figure 4 and Appendix A). And only compounds **6** and **7** had weak cytotoxic activities against tumor cell lines HCT116 and BT-20 with an IC_50_ of 18.2 μM and 11.3 μM, respectively (Appendix A).

### 2.3. Direct Cloning and Heterologous Expression of the gla BGC

Heterologous expression is also an effective strategy for mining and engineering the natural products. Herein, the *gla* BGC (76 kb) was cloned from the genome of ATCC 10248 via direct cloning based on the ExoCET method [40]. A potent constitutive promoter P_Genta_ was inserted in front of the gene *glaP* to drive the *gla* BGC expression, and the cassette *amp-attP* was also added into the *gla* expression vector to transfer the BGC into the specific site on the chromosome of the heterologous host via site-specific integration [41] (Figure 5a and Appendix A). The final construct of the BGC was electroporated into the heterologous host *C. brevitalea* DSM 7029 and conjugated into *B. thailandensis* E264 to investigate its products, respectively [29,33,42] (Appendix A).

The metabolite profiles showed that DSM 7029 and E264 carrying engineered *gla* BGC produced a series of evident peaks compared to the negative control DSM 7029 and E264 wild type. The peaks were target gladiofungin and its derivatives, which were confirmed via HPLC-MS/MS. The production yield of gladiofungins in the E264 was higher than that of the DSM 7029 (Figure 5b). Thus, heterologous expression was another suitable approach for mining the products biosynthesized by *gla* BGC, and DSM 7029 and E264 were appropriate heterologous hosts for the *gla* BGC and could express other exogenous BGCs from proteobacteria to discover new natural products.

## 3. Discussion and Conclusions

Glutarimide antibiotics characterized by a glutarimide ring were derived mainly from *Streptomyces* in the past but were recently found in various strains, including *Burkholderia*, *Pseudomonas*, and other marine-derived strains [9,10,11,12,13]. These compounds exhibit widespread pharmacological effects, such as antitumor, anti-inflammatory, and antifungal. The well-known members of this class of polyketides, iso-migrastatin (iso-MGS), lactimidomycin (LTM), cycloheximide (CHX), dorrigocins (DGN), and 9-methylstreptimidone, are best known as inhibitors of eukaryotic protein translation that have served as antitumor drug leads [43,44]. The cysteine adducts of iso-MGS, NK30424A, and NK30424B could also inhibit the PLS-induced TNF-α production by suppressing the NF-kB signaling pathway, while 9-methylstreptimidone could inhibit NO production and iNOS expression in LPS-stimulated RAW264.7 cells [45,46]. Like 9-methylstreptimidone, we found glidofungin derivatives (**2** and **4**) from *Burkholderia gladioli* ATCC 10248 also had significant anti-inflammatory activities, which showed the potential for the development of leads for anti-inflammatory agents. Apart from antitumor and anti-inflammatory activity, some of them, such as methylstreptimidone analogs and glidofungin A, have outstanding antibacterial and antifungal properties [10,47]. The discovery of more glutarimide antibiotics will provide a fertile source for understanding the potential mechanisms of their diverse biological activities.

Due to its complex structures and outstanding biological activity, the biosynthesis of glutarimide-containing polyketides has been well invested. The feature of the glutarimide moiety is formed from the conserved module block that benefits the genomic mining of this family of compounds in various strains, for example, the genome mining of gladiofungins and gladiostatins [10,11]. The structural diversity of this family is derived not only from different lengths of core carbon chains and complex post-modification reactions by comparing their structures but, most importantly, from variation chain termination mechanisms involved in terminal modules, such as TE and noncanonical AfsA-domain-mediated formation of the unusual butenolide moiety [9,10,11,18,48]. The butenolide group of gladiofungin A was formed from the unusual AfsA domain appending to the termination of *gla* BGC, which is known to catalyze the C3 precursors such as 3-glyceraldehyde phosphate (3-GAP) and DHAP derived from glycolysis to form butyrolactone or butenolide in *Streptomyces* [10,11,49]. In our work, seven gladiofungins derivatives with different moieties at the termination were isolated in the activated mutant of *gla* BGC through the potent promoter insertion. These new gladiofungins most likely originated from the spontaneous ring opening and decarboxylation of butenolide group in gladiofungin A (**1**) followed by a series of rearrangement reactions during the fermentation and isolation process as supported in previous studies, such as the isomerization of butenolide group and the degradation of gladiostatin [11,50].

The *gla* BGC was flanked by transposase genes that seemed to be a result of a horizontal gene transfer (HGT) in the complex ecology environment [36,37]. HGT is a major contributor to the genetic diversity in bacteria and essential for the evolution of social behavior in microbes [51,52]. The chemical diversification in the glutarimide family might be shaped by natural selection to fulfill functional roles in native environments through a variety of strategies, as 9-methylstreptimidone could employ module iteration, while cycloheximide might be through tailoring modification [43]. The unusual terminal module arrangement of *gla* BGC in *Burkholderia gladioli* strains is an AfsA domain that might result from the extensive genetic exchange. Interestingly, the Handelsman group identified a hybrid nonribosomal peptide synthetase and polyketide synthase containing a conserved module block of glutarimide moiety in the genome of *Kitasatospora mediocidica* [43]. Additionally, the biosynthesis of sesbanimide is terminated by the termination module of NRPS [12,13]. These findings extend the chemical and genetic diversity of this family of compounds that have more potential to be mined depending on the DNA sequencing. The *gla* BGC of model strain *B. gladioli* ATCC 10248 was silent under our standard laboratory conditions but was successfully activated by promoter insertion in this study. However, this family BGC from other non-model *B. gladioli* strains was expressed in a minimal salt medium supplemented with glucose [10,11]. Compared with the expression condition in non-model strains, we deduced that *gla* BGC was silenced in the model strain *B. gladioli* ATCC 1028, perhaps due to the lack of some important regulators or activators [53].

In conclusion, the silent *trans*-AT *gla* BGC in *B. gladioli* ATCC 10248 was successfully activated through two pathway-specific activation strategies of the in situ promoter insertion and the heterologous expression in this work. Seven corresponding glutarimide-containing polyketides, including five new ones, gladiofungins D-H (**3**–**7**), were elucidated through analysis of HR-ESI-MS and NMR spectroscopy. Compounds **2** and **4** had significant anti-inflammatory activities by inhibiting the NO production in LPS-induced RAW 264.7 macrophages, while compounds **6** and **7** had weak activities against tumor cell lines HCT116 and BT-20 with an IC_50_ of 18.2 μM and 11.3 μM, respectively. These new findings not only increase the structural diversities of glutarimide-containing compounds but also expand new structural types for the discovery of anti-inflammatory lead compounds. Furthermore, the heterologous expression of gladiofungin provides a potentiality for engineering the functional glutarimide and butenolide moieties into the other polyketides to improve the glutarimide-containing compound’s structural diversities and biological activities.

## Figures and Tables

**Figure 1 molecules-28-06937-f001:**
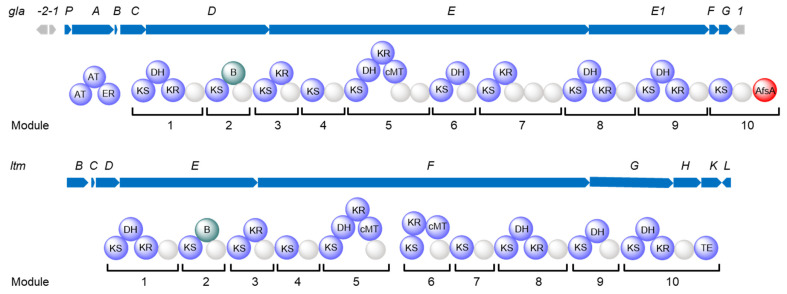
The comparison of the BGCs of gladiofungin BGC (*gla*) from *B. gladioli* ATCC 10248 and lactimidomycin BGC (*ltm*) from *Streptomyces* species.

**Figure 2 molecules-28-06937-f002:**
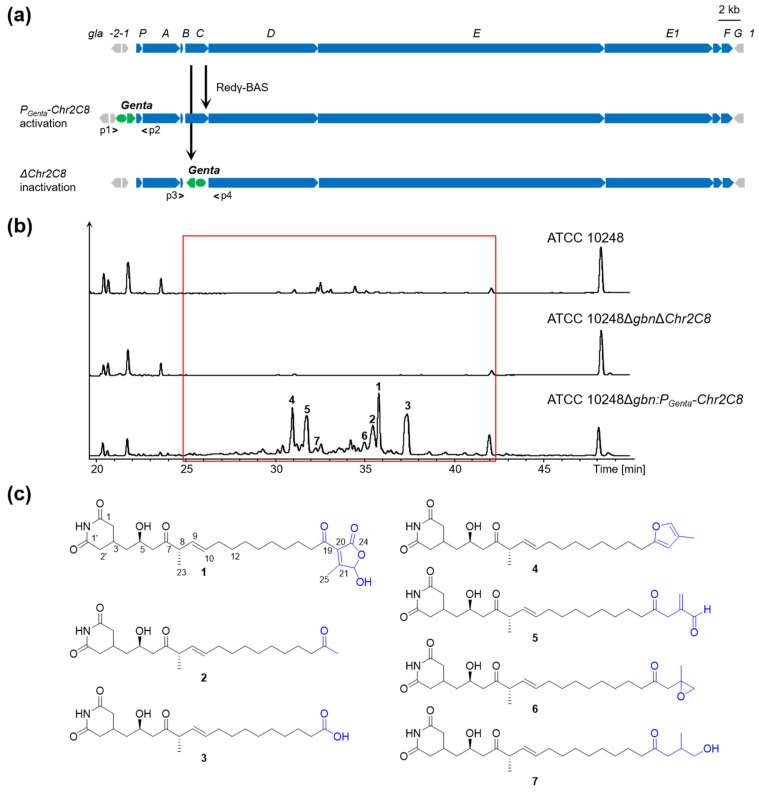
Promoter insertion directed discovery of gladiofungin derivatives from BGC 8 on chromosome 2 (Chr2C8) in *Burkholderia gladioli* ATCC 10248. (**a**) Diagram for construction of BGC 8 activation (P_Genta_-Chr2C8) and inactivation (ΔChr2C8) using Redγ-BAS recombinases. (**b**) HPLC-MS analysis of crude extracts from wild-type strain ATCC 10248 and mutants. Black indicates UV spectrum. Red box indicates different peaks. (**c**) Structures of purified compounds **1**–**7**.

**Figure 3 molecules-28-06937-f003:**
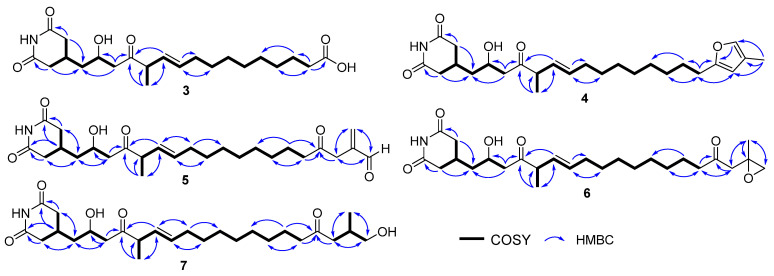
Key COSY and HMBC correlations of compounds **3**–**7**.

**Figure 4 molecules-28-06937-f004:**
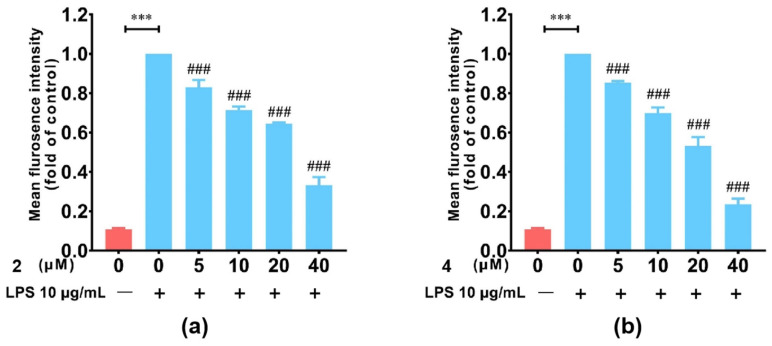
Inhibition of compounds **2** and **4** on LPS-induced NO production in RAW 264.7 macrophages. (**a**) Inhibition of **2** on LPS-induced NO production. (**b**) Inhibition of **4** on LPS-induced NO production. The negative control is without compounds and LPS. The other control is without compounds and with 10 μg/mL LPS, *** *p* < 0.001 (the LPS group is significantly different compared to the control group), ### *p* < 0.001 (the compound group is significantly different compared to the LPS group).

**Figure 5 molecules-28-06937-f005:**
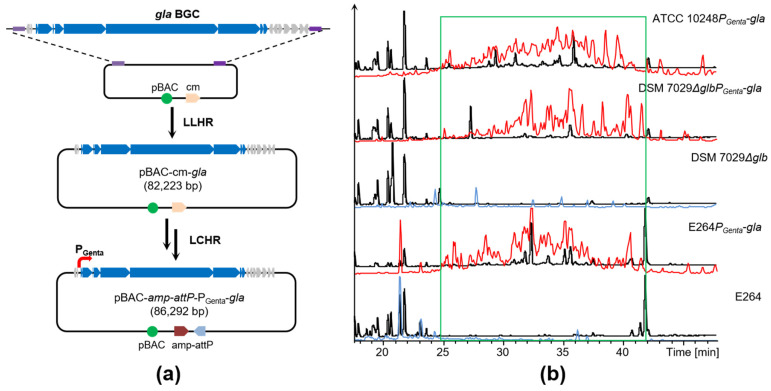
Heterologous expression of *gla* BGC. (**a**) A diagram of direct cloning and modification of *gla* BGC. LLHR: linear and linear homologous recombination; LCHR: linear and circle homologous recombination. The blue color indicates *gla* BGC; the grey color indicates additional genes. The purple color indicates homologous arms. (**b**) HPLC-MS analysis of crude extracts of wild-type strains and recombinants carrying *gla* BGC. Black color indicates UV spectrum; red and blue colors indicate BPC *m*/*z* 350–550; green box indicates target derivatives.

**Table 1 molecules-28-06937-t001:** The ^1^H (600 MHz) data of **3**–**7** in CDCl_3_.

No	3	4	5	6	7
2a	2.77, m *^a^*	2.76, m *^a^*	2.76, m *^a^*	2.77, m *^a^*	2.76, m *^a^*
2b	2.33, m *^a^*	2.33, m *^a^*	2.32, m *^a^*	2.33, m *^a^*	2.33, m *^a^*
3	2.49, m	2.47, m	2.48, m	2.46, m *^a^*	2.48, m
4a	1.60, m	1.60, m *^a^*	1.60, m *^a^*	1.58, m *^a^*	1.60, m
4b	1.34, m	1.33, m *^a^*	1.34, m *^a^*	1.34, m	1.34, m *^a^*
5	4.10, m	4.09, m	4.09, m	4.09, m	4.09, m
6a	2.65, dd (8.8, 18.1)	2.62, dd (8.8, 18.1)	2.63, dd (8.5, 18.1)	2.63, dd (8.7, 18.0)	2.63, dd (8.7, 18.1)
6b	2.57, dd (2.8, 18.1)	2.58, dd (2.8, 18.1)	2.57, dd (3.1, 18.1)	2.55, dd (2.7, 18.0)	2.57, dd (2.9, 18.1)
8	3.12, m	3.11, m	3.11, m	3.12, m	3.12, m
9	5.32, dd (8.3, 15.3)	5.32, dd (8.3, 15.3)	5.32, dd (8.4, 15.3)	5.32, dd (8.3, 15.2)	5.32, dd (8.4, 15.3)
10	5.59, dt (6.8, 15.3)	5.59, dt (6.7, 15.3)	5.58, dt (6.8, 15.3)	5.58, dt (6.7, 15.2)	5.59, dt (6.8, 15.3)
11	2.01, m	2.00, m	2.00, m	2.00, m	2.01, m
12	1.27, m *^a^*	1.33, m *^a^*	1.34, m *^a^*	1.26, m *^a^*	1.34, m *^a^*
13	1.27, m *^a^*	1.27, m *^a^*	1.26, m *^a^*	1.26, m *^a^*	1.26, m *^a^*
14	1.27, m *^a^*	1.27, m *^a^*	1.26, m *^a^*	1.26, m *^a^*	1.26, m *^a^*
15	1.27, m *^a^*	1.27, m *^a^*	1.26, m *^a^*	1.26, m *^a^*	1.26, m *^a^*
16	1.27, m *^a^*	1.27, m *^a^*	1.26, m *^a^*	1.26, m *^a^*	1.26, m *^a^*
17	1.62, m	1.60, m *^a^*	1.60, m *^a^*	1.58, m *^a^*	1.55, m
18	2.33, t (7.4)	2.54, t (7.5)	2.48, t (7.4)	2.46, m *^a^*	2.41, m
20a	1.15, d (6.9)	5.83, s	3.35, s	2.79, d (16.6)	2.53, m
20b				2.54, d (16.6)	2.33, m *^a^*
21					2.21, m
22a		7.04, s	9.52, s	3.45, d (11.2)	3.55, m
22b				3.37, d (11.2)	3.83, m
23		1.14, d (6.8)	1.14, d (6.9)	1.14, d (6.8)	1.15, d (6.8)
24a		1.97, s	6.37, s	1.19, s	0.92, d (7.0)
24b			6.22, s		
2′a	2.77, m *^a^*	2.76, m *^a^*	2.76, m *^a^*	2.77, m *^a^*	2.76, m *^a^*
2′b	2.33, m *^a^*	2.33, m *^a^*	2.32, m *^a^*	2.33, m *^a^*	2.33, m *^a^*
1-NH	8.36, s	8.21, s	8.08, s	8.29, s	8.04, s

*^a^* Overlapped.

**Table 2 molecules-28-06937-t002:** The ^13^C (150 MHz) Data of **3**–**7** in CDCl_3_.

No	3	4	5	6	7
1	172.7, C	172.4, C	172.3, C	172.5, C	172.3, C
2	38.5, CH_2_	38.5, CH_2_	38.5, CH_2_	38.5, CH_2_	38.5, CH_2_
3	27.2, CH	27.2, CH	27.2, CH	27.2, CH	27.2, CH
4	40.8, CH_2_	40.8, CH_2_	40.9, CH_2_	40.8, CH_2_	40.8, CH_2_
5	65.0, CH	64.9, CH	64.9, CH	64.9, CH	64.9, CH
6	47.2, CH_2_	47.2, CH_2_	47.1, CH_2_	47.2, CH_2_	47.1, CH_2_
7	213.4, C	213.4, C	213.3, C	213.3, C	213.4, C
8	51.0, CH	51.0, CH	51.0, CH	51.0, CH	51.1, CH
9	128.4, CH	128.3, CH	128.3, CH	128.4, CH	128.4, CH
10	134.6, CH	134.7, CH	134.6, CH	134.6, CH	134.7, CH
11	32.6, CH_2_	32.7, CH_2_	32.6, CH_2_	32.6, CH_2_	32.6, CH_2_
12	29.2, CH_2_	29.3, CH_2_	29.2, CH_2_	29.4, CH_2_	29.1, CH_2_
13	29.0, CH_2_	29.4, CH_2_	29.2, CH_2_	29.1, CH_2_	29.1, CH_2_
14	29.2, CH_2_	29.4, CH_2_	29.4, CH_2_	29.3, CH_2_	29.3, CH_2_
15	29.2, CH_2_	29.3, CH_2_	29.3, CH_2_	29.2, CH_2_	29.3, CH_2_
16	29.0, CH_2_	29.2, CH_2_	29.1, CH_2_	29.2, CH_2_	29.3, CH_2_
17	24.8, CH_2_	28.2, CH_2_	23.8, CH_2_	23.5, CH_2_	23.9, CH_2_
18	33.9, CH_2_	28.2, CH_2_	43.1, CH_2_	44.9, CH_2_	43.6, CH_2_
19	178.4, C	156.7, C	206.9, C	214.3, C	212.1, C
20	16.0, CH_3_	107.6, CH	41.7, CH_2_	48.3, CH_2_	47.3, CH_2_
21		120.5, C	143.5, C	72.6, C	32.1, CH
22		137.3, CH	193.6, CH	70.0, CH_2_	68.1, CH_2_
23		16.1, CH_3_	16.0, CH_3_	16.0, CH_3_	16.0, CH_3_
24		9.9, CH_3_	137.1, CH_2_	24.5, CH_3_	17.1, CH_3_
1′	172.5, C	172.3, C	172.3, C	172.4, C	172.2, C
2′	37.2, CH_2_	37.3, CH_2_	37.3, CH_2_	37.3, CH_2_	37.3, CH_2_

## Data Availability

The data supporting the findings of this study are available within the article and the Appendix A.

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
