# Peer review of "The Genomic-Driven Discovery of Glutarimide-Containing Derivatives from Burkholderia gladioli"

_molecules, 2023, doi:10.3390/molecules28196937_

Round 1
Reviewer 1 Report
Bian and Zhou et al submitted a manuscript entitled “Genomics-Directed Discovery of Glutarimide-Containing Derivatives from Burkholderia gladioli”, in which they talked about new glutarimide-containing derivatives identified with the assistance of genomic study. Generally, this manuscript included a large amount of data, especially for the amazing part of structural elucidation of these derivatives. But I have some concerns regarding this manuscript and the authors may need to re-organize this manuscript.
One of a major concern of mine is these compounds are genuine natural products or degradation products from some known compound. As mentioned in ref. 9 of this manuscript (Angew. Chem. Int. Ed. 2020, 59, 23145 – 23153), compound 1 is not stable and prone to degrade to compound 2 during growth periods. It made me doubt if other compounds are also degradation intermediates/products of compound 1, considering the structural similarity of these compounds. And the authors did observe instability during purification (line 337)
To the best of remove this degradation possibility, some suggestions:
a) Improve the description of part 4.6 in page 10, especially for the extraction and isolation part. Some details can be included:
For line 322-336
Was the fermentation process involved with extra supplement of oxygen or just in the air (Oxygen-mediated degradation)?
Was light avoided during fermentation (Light-mediated degradation)?
What was the temperature used during concentration of EtOAc (Temperature-mediated degradation)?
In MPLC/semi-preparative HPLC, is there any additive used in mobile phase? What was the column temperature (pH-mediated degradation)?
Instability was observed during purification, did the author test the degradation mixture with LCMS?
For line 337-350:
0.1% TFA was used in semi-preparative HPLC, did the authors check and find any degradation during purification?
b) If the authors still have compound 1 in hand, check if these compounds have degraded. If yes, test the degradation mixture with LCMS to see any known compounds could be identified. If no, it is recommend running a stability check, especially for compound 1.
Author Response
Thank you very much for taking the time to review our manuscript.
Please find our point-by-point response attached.

Reviewer 2 Report
The manuscript titled “Genomics-Directed Discovery of Glutarimide-Containing Derivatives from Burkholderia gladioli” presents an intriguing study on the genomic-directed discovery of glutarimide-containing derivatives from Burkholderia gladioli. The approach of activating a silent biosynthetic gene cluster to produce these derivatives is innovative and holds promise for the discovery of other natural products. The study effectively combines bioinformatics, genomics, and organic chemistry to elucidate the production of glutarimide-containing derivatives. The methodology, particularly the activation of the silent gene cluster, is well-executed and could serve as a model for similar studies. The discovery of five new glutarimide-containing analogues is a significant contribution to the field.
Some minor comments:
1. Anti-inflammatory activities for other compounds (3, 5, 6, and 7) should be depicted as well, at least in the Supplementary Figure.
2. The potential applications and significance of the discovered compounds should be discussed in greater depth. Comparisons with existing compounds and potential mechanisms of action would enhance the discussion.
3. The evolutionary significance of the silent gene cluster in Burkholderia gladioli could be discussed. Why is this cluster silent, and what are the evolutionary pressures that led to this?
In conclusion, the manuscript is well-written and presents a novel approach to discovering new natural products. I recommend accepting the manuscript after minor revisions addressing the above-mentioned points.
Author Response

(The authors gave the same response as above.)

Reviewer 3 Report
1. The methods about bioinformatic analysis used in this manuscript can be added.
2. Section 2.1 Line 114 to 115, reference(s) can be added for this sentence.
3. For figure 4, the explanation for (a) and (b) can be added in the legend.
4. In Discussion section, Line 212 to 214, reference(s) can be added for this sentence.
5. In Discussion section, some more examples about bioinformatic analysis related with glutarimide can be added.
Author Response

(The authors gave the same response as above.)

Round 2
Reviewer 1 Report
The authors have given satisfactory replies to my concerns. No further issues need to be addressed. It is an excellent work!